# Literature Review: Evaluation of Drug Removal Techniques in Municipal and Hospital Wastewater

**DOI:** 10.3390/ijerph192013105

**Published:** 2022-10-12

**Authors:** Henry Rodríguez-Serin, Auria Gamez-Jara, Magaly De La Cruz-Noriega, Segundo Rojas-Flores, Magda Rodriguez-Yupanqui, Moises Gallozzo Cardenas, José Cruz-Monzon

**Affiliations:** 1Escuela de Ingeniería Ambiental, Facultad de Ingeniería, Universidad Cesar Vallejo, Trujillo 13007, Peru; 2Vicerrectorado de Investigación, Universidad Autónoma del Perú, Lima 15842, Peru; 3Departamento de Ciencias, Universidad Privada del Norte, Trujillo 13007, Peru; 4Universidad Tecnológica del Perú, Trujillo 13011, Peru; 5Facultad de Ingeniería Química, Universidad Nacional de Trujillo, Av. Juan Pablo II, Trujillo 13011, Peru

**Keywords:** technique, removal, pharmaceuticals, wastewater, efficiency

## Abstract

There are several techniques for the removal of pharmaceuticals (drugs) from wastewater; however, strengths and weaknesses have been observed in their elimination processes that limit their applicability. Therefore, we aimed to evaluate the best techniques for the removal of pharmaceuticals from municipal and hospital wastewater. For this, a non-experimental, descriptive, qualitative–quantitative design was used, corresponding to a systematic review without meta-analysis. Based on established inclusion and exclusion criteria, 31 open-access articles were selected from the Scopus, ProQuest, EBSCOhost, and ScienceDirect databases. The results showed that high concentrations of analgesics such as naproxen (1.37 mg/L) and antibiotics such as norfloxacin (0.561 mg/L) are frequently found in wastewater and that techniques such as reverse osmosis, ozonation, and activated sludge have the best removal efficiency, achieving values of 99%. It was concluded that reverse osmosis is one of the most efficient techniques for eliminating ofloxacin, sulfamethoxazole, carbamazepine, and diclofenac from municipal wastewater, with removal rates ranging from 96 to 99.9%, while for hospital wastewater the activated sludge technique proved to be efficient, eliminating analgesics and antibiotics in the range of 41–99%.

## 1. Introduction

In recent years, analytical techniques have been developed that allow the recognition and quantification of the presence of many pollutants—and even micropollutants—at low concentrations (from ng/L to µg/L) in water [1,2]. Among them are the so-called emerging contaminants (ECs), whose presence in and possible effects on water bodies cause special concern [3]. According to Jari et al. (2022), these emerging contaminants are not fully regulated; however, they are used in large quantities, which favors their presence in wastewater and surface water; for these reasons, they can be harmful to aquatic ecosystems and human health [4].

Similarly, Morosini et al. (2017) indicated that this group includes drugs and pharmaceutical compounds (PhCs)—products that have been widely studied in recent years due to their presence in the aquatic environment and their negative impacts [5]. In addition, the attention they receive is mainly due to their high consumption, production, improper disposal, bioaccumulation, and non-biodegradable nature [6]. According to Silva et al. (2020), drugs are metabolized and excreted after human ingestion, leading to a mixed discharge of active ingredients and metabolites into the sewage networks [7]. In the same line, Zhang et al. (2016) indicated that PhCs (pharmaceuticals) enter the environment mainly through discharges from wastewater treatment plants (WWTPs), hospital effluents, industrial effluents, runoff, and human and animal excreta [8]. Similarly, Krakkó et al. (2019) argued that the main sources of contamination by pharmaceuticals are related to hospital and municipal WWTP effluents, with hospital effluents containing higher concentrations of certain PhCs [9,10]. In addition, due to the COVID-19 pandemic, the consumption of pharmaceutical products has increased worldwide, raising their concentrations in wastewater [11]. Regarding the above, Krishnan et al. (2021) stated that traditional wastewater treatment methods are not designed to remove PhCs, causing effluents to be discharged into water bodies without being efficiently treated. This causes negative impacts on biodiversity and favors resistance to antibiotics in some bacteria [12]. Therefore, several techniques are currently applied for the elimination of drugs, based on physical, chemical, and biological methods [13].

In this sense, each treatment technique has advantages and limitations; for example, treatment techniques based on physical methods such as adsorption and membrane filtration have been proven to remove drugs effectively. However, adsorption has the limitation of only transferring pollutants from the water to a porous structure, resulting in residues that will later require treatment or final disposal [14]. Likewise, membrane filtration faces challenges such as high operational energy demand [15]. Techniques based on chemical methods such as ozonation and photocatalysis have also been shown to be effective in removing drugs [16]. However, ozonation can generate harmful byproducts, and photocatalysis is one of the most investigated techniques, but its energy demand and its operational and maintenance costs have prevented large-scale research to date [17]. On the other hand, techniques based on biological methods are usually less expensive, but their removal of some pharmaceuticals is deficient; for example, the removal of pharmaceuticals by activated sludge will depend on the characteristics of these pollutants, as well as on the conditions under which the operation takes place [18]. Another biological technique with promising results in pharmaceutical drug removal is membrane bioreactors; however, their large-scale application has limitations, such as membrane fouling and operating costs [19].

Similarly, the use of white-rot fungi and their oxidoreductase enzymes to remove drugs has few reports in non-sterile conditions and real wastewater, and large-scale implementation of these techniques requires overcoming problems such as the addition of complementary substrates, partial re-inoculation of fresh fungi, or the use of extended hydraulic retention times [20]. It is worth mentioning that fungal treatment has shown very promising potential in drug removal; however, discussion of these techniques is beyond the scope of this review, so readers are encouraged to consider these approaches in drug removal as well. This review focuses on data from studies that were conducted using real, non-synthetic wastewater in order to provide a picture of real systems. Additionally, hybrid treatments that integrate two or more treatment methods (i.e., physical, chemical, and/or biological) to facilitate pollutant removal [21] were left out of the analysis, with the objective of evaluating the removal efficiency of each method separately.

The main objective of this review is to show the best techniques for removing pharmaceuticals from municipal and hospital wastewater, due to the worldwide increase in the consumption of pharmaceuticals and their concentrations in wastewater. Taking into account that conventional treatments used in WWTPs do not efficiently remove PhCs, causing effluents to be discharged with high contents of these compounds in different water bodies, there is a need to identify and evaluate new techniques to solve this problem. At the same time, we provide information related to these techniques, which can help to avoid negative impacts on aquatic biodiversity and human health. In addition, this research serves as a basis for future research to carry out experimental investigations.

## 2. Materials and Methods

This was a basic, descriptive, documentary-type research study whose bibliographic information was considered within the period 2016–2022, based on the evaluation of the best techniques for removing pharmaceuticals from municipal and hospital wastewater, establishing the types of pharmaceuticals present in municipal and hospital wastewater, and evaluating the percentage of pharmaceutical removal according to the treatment methods applied and the type of wastewater. An a priori categorization matrix was elaborated for this research, considering the categories and subcategories shown in Figure 1. 

### 2.1. Data Collection Techniques and Instruments

The research was conducted using the Scopus, ProQuest, EBSCOhost, and ScienceDirect databases, considering original articles from internationally indexed journals that met the inclusion and exclusion criteria of the topic. The analysis technique was based on the work of Marín-González et al. (2018), because it encompasses analytical–synthetic processing, which includes bibliographic description, classification, extraction, and preparation of reviews [22].

### 2.2. Procedure

In this research, the scientific articles related to the studied topic were identified using keywords and their respective combinations through Boolean operators, as shown in Figure 2, considering open-access scientific articles published in English from 2016 to 2022. Likewise, the use of synthetic wastewater as a matrix and the use of hybrid techniques for the elimination of drugs in the investigations were considered exclusion criteria for the present systematic review. Subsequently, the articles that met the aforementioned criteria were organized according to the research design used in the publications, separating publications with a pre-experimental design with no strict control of variables, corresponding to studies conducted in the field; those with a quasi-experimental design where some variables are controlled, understood as those studies conducted at a pilot level; and those with a pure experimental design where all variables are controlled, which typically correspond to studies at the laboratory level. In addition, 31 peer-reviewed articles from open-access indexed journals were considered. According to the Scimago Journal & Country Rank (SJR) website, as of 2021, 74.2% of the indexed journals analyzed belonged to the Q1 quartile, and 25.8% belonged to the Q2 quartile (see Appendix A). The data collection form can be seen in Table 1, which contains the information necessary to obtain the research results.

On the other hand, Figure 2 shows the schematization of the selection process for the scientific articles included in this research. This process allowed the compilation of a total of 468 publications, eliminating 54 duplicated publications (414 remaining publications); then, 86 publications corresponding to book chapters and review articles were excluded (328 remaining), and then a further 174 publications were eliminated because, after analyzing their abstracts, it was evident that they would not help in the development of the objectives (154 remaining). Subsequently, the remaining scientific articles were subjected to an analysis applying the established exclusion criteria, and 123 scientific articles were excluded. Finally, the remaining 31 scientific articles were included for analysis and critical evaluation in this study. The 31 studies included in this research were conducted in countries such as Spain (8), China (3), Portugal (2), Nigeria (2), Cyprus (1), the United States (1), Turkey (1), Colombia (1), Sweden (1), India (1), South Korea (1), Finland (1), Greece (1), South Africa (1), Denmark (1), Saudi Arabia (1), Kenya (1), Slovakia (1), Iran (1), and Mexico (1).

## 3. Results

The trends of publications related to the techniques for removing pharmaceuticals from municipal wastewater (MWW) and hospital wastewater (HWW) were identified, with 290 publications on MWW and 178 publications on HWW, showing a lineal increase in recent years (Figure 3). On the other hand, the results observed in Figure 3 are consistent with the findings of Alvarino et al. (2018) in their review article on trends in the removal of micropollutants from wastewater, in which the authors mentioned that drug contamination was rarely investigated by scientists prior to 2015. Moreover, in those years, there was no highly sensitive analytical equipment to deal with these cases in many countries [54]. Thus, it can also be deduced that the removal of pharmaceuticals from wastewater is currently a relevant topic for the scientific community. According to Kalaboka et al. (2020), in addition to classical pollutants, the appearance of emerging contaminants such as pharmaceuticals has attracted increasing interest in the field of environmental research, since these pollutants have been frequently detected in different aquatic environments. [55] At the same time, the interest from part of the scientific community in searching for efficient techniques to remove pharmaceuticals from hospital wastewater before its discharge into sewers or water bodies has greatly increased in recent years [56,57].

On the other hand, in wastewater, the most detected and studied PhCs belong to therapeutic classes such as antibiotics, analgesics, hormones, and antiepileptics. Additionally, lipid regulators, beta-blockers, radiocontrast agents, and psychotropics have been studied, but to a lesser extent [33]. Although there a large number of PhCs are detected in wastewater, this review considers only those most frequently detected at concentrations that may represent a threat to the environment according to the literature. Based on the aforementioned criteria, 16 PhCs were selected as the main focus of this research: 6 antibiotics (i.e., ciprofloxacin (CIP), norfloxacin (NOR), ofloxacin (OFL), enrofloxacin (ENR), sulfamethoxazole (SMX), and sulfadiazine (SDZ)); 1 antiepileptic (i.e., carbamazepine (CBZ)); 5 analgesics (i.e., paracetamol or acetaminophen (ACE), diclofenac (DCF), ibuprofen (IBU), naproxen (NPX), and ketoprofen (KET)); and 4 hormones (i.e., estrone (E1), 17-β-estradiol (E2), estriol (E3), and 17α-ethinylestradiol (EE2)). Below is the analysis performed to establish the types of pharmaceutical drugs present in MWW and HWW. It should be noted that the data were taken only from the research articles shown in Appendix A.

Table 2 and Table 3 show the presence of pharmaceuticals in wastewater, where antibiotics have been evidenced in several studies. These are commonly used PhCs that protect animals and humans from diseases and infections caused by bacteria [58]. Likewise, Cristóvão et al. (2020) emphasized that special interest should be given to antibiotics since they are widely consumed drugs, persist in wastewater treatment, and facilitate the development of antibiotic-resistant bacteria, which may cause harmful effects on human and environmental health [39]. In this study, the highest antibiotic concentrations in MWW reported in the studies reviewed ranged from 27 to 4200 ng/L for CIP and from 12 to 7800 ng/L for SMX. As for HWW, the concentration ranges were 70–561,000 ng/L for NOR, 124,000–198,000 ng/L for OFL, and 120–120,000 ng/L for CIP. In general, all antibiotic concentrations in HWW were higher than in MWW, with CIP and SMX being the most frequently detected drugs (Figure 4a). These results were consistent with previous studies that showed that the concentrations of antibiotics in HWW were 3–10 times higher than those in MWW [37,59]. In the same context, Kutuzova et al. (2021) noted that the most frequently detected antibiotics in wastewater were SMX and CIP [60].

Similarly, the class of antiepileptic drugs—also known as anticonvulsants—comprises drugs that are often used to treat people with epilepsy or other particular mental disorders [61]. As shown in Table 2 and Table 3, the concentrations of the antiepileptic CBZ ranged from 96 ng/L to 6500 ng/L in MWW and from 41.1 ng/L to 880 ng/L in HWW, representing the third most frequently detected class of pharmaceutical (Figure 4a). Concerning the above, Nkoom et al. (2019) argued that CBZ is the most studied antiepileptic drug by the scientific community due to its low biodegradability, high consumption, and low removal efficiency in wastewater treatment plants [62].

Another class of pharmaceuticals present in wastewater is analgesics, which are used for pain relief and are considered to be important environmental pollutants [58]. The data shown in Table 2 and Table 3 evidence high concentrations of all analgesic drugs, with the highest values for contaminants such as NPX (60–1,370,000 ng/L), ACE (156.4–623,000 ng/L), and DCF (6–410,000 ng/L) in MWW. Regarding HWW, the contaminants with the highest concentrations were DCF (590–166,000 ng/L) and ACE (2660–119,500 ng/L). In addition, all analgesics were more concentrated in MWW, where they represented the most frequently detected class of pharmaceuticals (Figure 4b). The results found were consistent with the findings of Kermia et al. (2016), who noted that analgesics are the most frequently detected pharmaceuticals in wastewater due to their high consumption and the fact that they are sold without a medical prescription [63]. Hormonal drug classes are known because they may cause endocrine disorders and affect the sexual and reproductive systems of species such as fish [58,64]. In this research, the ranges of influent concentrations of E1, E2, and E3 were 78–158, 11–54, and 42-162 ng/L, respectively, in MWW. In HWW, the influent concentrations of E1 and E2 were similar to the data found for MWW; however, higher concentrations were evidenced for E3 (27–1480 ng/L) and EE2 (881–9833 ng/L). In this type of wastewater, hormones were the least frequently detected drug class (Figure 4b) and had the lowest concentrations compared to the other compounds. These findings were similar to those obtained by another author [65].

The data in Table 4 show the percentages of pharmaceutical removal in terms of removal techniques based on physical treatment methods, where reverse osmosis stands out, with removal rates ranging from 96% to 99.9% for pharmaceutical products such as OFL, SMX, CBZ, and DCF in studies with a pre-experimental design [48]. On the other hand, Farrokh Shad et al. (2019), in a study with a quasi-experimental design, reaffirmed the high removal of this technique regarding the aforementioned contaminants, along with the complete removal of hormones such as E1 and E2 [29].

Additionally, in research with a quasi-experimental design, the nanofiltration technique showed removal efficiency similar to that of reverse osmosis. Garcia-Ivars et al. (2017), in their study, managed to remove contaminants in the range of 85% to 99% for pharmaceutical drugs such as SMX, ACE, DCF, IBU, and NPX. In this sense, the authors indicated that removal efficiency was related to the membrane pore size, demonstrating higher efficiency of removal in membranes with the smallest pore size [27], as was also identified in previous research [66,67]. According to Azizi et al. (2022), reverse osmosis uses a smaller pore size than nanofiltration, making it more efficient; however, because nanofiltration has a lower long-term cost overrun, it is commonly considered an appropriate technique as well [68].

For this research, adsorption by granular activated carbon showed low removal rates of antibiotics, antiepileptics, and most analgesics [31]. On the other hand, Sun et al. (2017), in a study with a pure experimental design, employed adsorption by powdered activated carbon for the removal of hormones, achieving removal rates of 34.4%, 83.3%, and 99. 9% for E1, E2, and EE2, respectively [49]. However, the applicability of these techniques at larger scales cannot be affirmed, since the literature indicates that adsorption is a practical process but it is mostly performed in studies with a pure experimental design, and there has been no evaluation of the cost of materials for large-scale operation [69].

Table 5 shows the removal of pharmaceuticals by chemical treatment methods. In this regard, no studies with a pre-experimental design were found in this research, and this is consistent with the findings of Benstoem et al. (2017), who mentioned that, as this is a relatively new topic, there are only studies using ozonation to remove pharmaceuticals on a large scale in countries with strict regulations, such as Switzerland [70]. Similarly, photocatalysis has not yet been evaluated on a large scale at present [17].

In studies with a quasi-experimental design, the technique with the highest removal percentages was ozonation, showing values of 90% for SMX and CBZ [25] and values above 95% for the hormones E1, E2, and EE2 [49]. In studies with a pure experimental design, the same technique positively eliminated the aforementioned pharmaceuticals and, in the study by Dogruel et al. (2020), even CBZ was eliminated, while pharmaceuticals such as DCF and NPX were eliminated by more than 96% [34]. In other studies, high percentages of drugs such as CBZ, SMX, and DCF have also been eliminated using different doses of ozone [71]. Regarding the heterogeneous TiO_2_ photocatalysis technique, all of the studies analyzed the elimination of only one or two drugs, enriching their initial concentrations (from 100,000 ng/L to 3,000,000 ng/L) [28,35,43,50]. In this regard, one study reported that this type of technique is more efficient when applied at higher concentrations [66]. Conversely, for Krishnan et al. (2021), this would represent a situation to be improved, because these initial concentrations are very high compared to those found in reality [12]. In this regard, in studies with a pure experimental design, Karaolia et al. (2018) removed 87% of SMX [28], while another study removed only 21% of the same drug [50]. This difference may have been because these studies were carried out in real wastewater, which may have different characteristics and other compounds present that also react with the radicals formed in the process [72].

According to Table 6, among the biological treatment methods, the activated sludge technique was the most widely used in the studies analyzed, which is consistent with the findings of several studies indicating that biological treatments are widely used and that activated sludge is the most widely used technique in the world [4,8,10]. This technique presented the highest number of pharmaceuticals to be eliminated, with very variable removal data. In studies with a pre-experimental design, this technique was able to remove antibiotics and analgesics at rates above 90% [24,26,33]. On the other hand, drugs such as the antiepileptic CBZ [23,33,41] and hormones [24] had low removal percentages—results that were consistent with those reviewed in other studies [73].

In the studies that used the membrane bioreactor technique, there were no remarkable removals except for the findings of Wang et al. (2018), who were able to eliminate hormones such as E1, E2, and E3 at rates of more than 82% during their study with a quasi-experimental design [30]. However, another study considered this technique to be more effective than that of activated sludge, so it should be further investigated [17].

Regarding the high-rate algal pond (HRAP) technique, it can be inferred that the three studies that employed this technique, using a quasi-experimental design, obtained low removal rates for all pharmaceuticals except for ACE (removal between 94.4 and 100%) [36,45,46]—results that were consistent with the findings of Vassalle et al. (2020) who, in their quasi-experimental study, concluded that HRAP was a natural treatment technique with low cost, which was viable for small populations and presented highly variable removal efficiencies [74].

In general, it was evident that these techniques were not entirely outstanding, the main reasons for which were detailed in the scientific literature, including the fact that they were not designed to eliminate these contaminants and that their efficiency greatly depends on the properties of each pharmaceutical drug—such as its biodegradability—as well as on the climatic conditions [6,41]. For example, in a study using the activated sludge technique, the removal of pharmaceuticals was over 73%, and this was attributed to the high levels of sunlight at the study site [26]. In contrast, in a study using the membrane bioreactor technique at a site with low temperatures, pharmaceuticals were only removed in the range of 25–38% [53]. In addition, Aydin et al. (2019) previously concluded that higher water temperatures can contribute to enhancing microbial biodegradation activity [75].

It should be noted that biological treatment techniques showed negative removal rates for drugs such as SMX, SDZ, and CBZ (Table 6), indicating that these micropollutants had higher concentrations in the effluents than in the influents; on this topic, Blair et al. (2015) argued that among the reasons for the negative removal is the fact that drugs may be enclosed in fecal particles, and when microbes break down these feces, the drugs are released [76]. Another reason for this phenomenon is that the metabolites of the drugs are transformed back into the original compounds by the action of microorganisms [77]. In addition, for Clara et al. (2004), negative removal rates were also attributed to daily fluctuations in concentration during the sampling period [78]. 

Finally, the analysis of the removal of pharmaceuticals from municipal wastewater is shown in Figure 5, Figure 6 and Figure 7; for hospital wastewater, it is shown in Table 7.

Considering Figure 5, the ozonation technique was the one that achieved the highest removal percentages in studies with a pure experimental design, since it eliminated SMX, CBZ, DCF, and NPX at rates of 86%, 100%, 98%, and 96%, respectively [34]. Similarly, the technique that stood out for the removal of the hormones E2 (83.3%) and EE2 (99.9%) was adsorption with powdered activated carbon [49].

On the other hand, Figure 6 shows that, in studies with a quasi-experimental design, the technique with the highest percentage of removal was reverse osmosis (from 59.3% to 100%). Furthermore, this technique eliminated all classes of pharmaceuticals included in this study [29]. Likewise, techniques such as ozonation and membrane bioreactors resulted in hormone removal rates greater than 95.7% and 82%, respectively [30,49].

Regarding Figure 7, in studies with a pre-experimental design, the reverse osmosis technique had higher removal percentages compared to the activated sludge and membrane bioreactor techniques for antibiotics and, in particular, for CBZ, since the latter was removed by 96.1% compared to the removal rates of −50 and 28% achieved by the studies using the activated sludge technique to remove the same compound [41,48]. However, the removal of analgesic drugs such as IBU and NPX was higher in the studies using activated sludge (higher than 89%) than in the studies using reverse osmosis (maximum values of 58.3%) [33,41,51].

Although studies with pure experimental and quasi-experimental designs indeed achieved significant levels of removal, it should be taken into account that the studies with a pre-experimental design had gone through a series of procedures to be implemented on a large scale or in the field, so it is feasible to discuss and contrast them [79]. Therefore, as mentioned above, large-scale wastewater treatments use primary treatment; secondary treatment (the most commonly used technique is activated sludge) and tertiary treatment (such as reverse osmosis, for example) are used if more purified wastewater is required to avoid contamination of water bodies or if the water is to be reused [41].

Therefore, several studies have shown that the activated sludge technique removes pharmaceuticals variably, and completely eliminates biodegradable pharmaceuticals such as ACE in municipal wastewater [50,80], but those that are more resistant to biodegradation—such as CBZ and SMX, among others—require a tertiary treatment for their elimination, such as reverse osmosis [48,80].

One of the reviewed studies concluded that reverse osmosis was efficient in the removal of a wide variety of pharmaceutical drugs; however, at the same time, it indicated that this technique had a deficiency due to the generation of a concentrated phase containing the drugs [41]. Regarding this deficiency, a review of membrane separation processes indicated that this concentrate should be sustainably managed and that advanced oxidation processes can be effectively applied to it [66]. However, according to De Ilurdoz et al. (2022), the advantages of this technique outweigh its disadvantages, since its large-scale application is effective in countries such as Spain and Croatia, demonstrating the complete removal of antibiotics [79].

Table 7 shows that the removal of pharmaceuticals from hospital wastewater has been investigated to a lesser extent, and this is because in many countries such wastewater is discharged into the sewers and treated together with municipal wastewater [8]. However, hospitals are considered to be a major source of pharmaceutical contamination, so treating them separately would be appropriate, as some pharmaceuticals may be present in high concentrations and would, therefore, be easier to treat [15,56].

Likewise, Table 7 shows that activated sludge removal techniques have been applied in studies with a pre-experimental design. Consequently, this technique resulted in the removal of antibiotics and analgesics in the range of 41% to 99% [23,24,26], although there was evidence of low removal of CBZ (range: −116 to −9%) and hormones, with very variable percentages [23,24]. From another perspective, techniques such as ozonation and photocatalysis have been investigated to remove pharmaceuticals from these waters, but the studies were conducted with a pure experimental design, so it cannot be asserted that the removal rates of over 86.6% achieved by these techniques would be the same in pre-experimental designs [25,43].

Tulashie et al. (2018) also reported that hospital effluent treatment studies have been conducted using techniques such as membrane filtration, activated carbon adsorption, and advanced oxidation processes. However, these techniques were found to be expensive and difficult to operate, especially in hospitals established in developing countries seeking to treat these effluents [15]. In contrast, techniques based on biological methods can produce effluents that preserve water quality standards at a reasonable cost for these facilities [41].

On the other hand, one study determined that membrane bioreactors are a viable technique to remove pharmaceuticals from hospital wastewater [58], but this study did not include articles that used this technique—possibly due to the exclusion criteria, which left out studies that used synthetic wastewater as a matrix. Incidentally, Taoufik et al. (2021) indicated that the membrane bioreactor technique shows greater removal of certain classes of pharmaceuticals than the activated sludge technique, but the difference was not extreme [17].

After reviewing the literature, it became evident that the physical methods evaluated show a recovery (i.e., removal of the drug from the aqueous matrix) of contaminants close to 100%, as is the case for reverse osmosis, so it would be convenient to propose an evaluation of the existing treatment methods for the recovered substances in order to fully complement their elimination. On the other hand, the evaluation of chemical and biological methods implies non-detection as an active principle, but it does not mean that the byproducts generated can potentially constitute equal or greater contaminants than the original active principle, as reported by Sun et al. (2021), who identified up to seven possible degradation products of tetracycline using *Phanerochaete chrysosporium*, where the transformation pathways included demethylation, dimethylamino oxidation, decarbonylation, hydroxylation, and oxidative dehydrogenation [81]. On the other hand, in the case of treatments using advanced oxidation processes, García-Galán et al. (2020) noted the possibility of potential ecotoxicity of the products generated during the transformation, which could generate a route of entry of substances into aquatic ecosystems and, consequently, problems of bioaccumulation and/or biomagnification [45]. The same authors evaluated the degradation of the antidepressant venlafaxine and its main metabolite O-desmethylvenlafaxine using advanced oxidation with UV/H_2_O_2_, removing about 99.9% of the compound, but at the same time identifying 11 and 6 transformation products, respectively [82].

Therefore, not only the presence or absence of the original substances, but also the possible metabolites generated and their possible synergistic effects that could be generated in the water bodies should be evaluated. In this sense, the research included in this review is not conclusive regarding the possible substances formed during the application of the treatment methods, so the conclusions on the efficiency of elimination are based exclusively on the capacity of the techniques to remove substances from the aqueous phase or to transform the bioactive species into others that are theoretically less harmful.

## 4. Conclusions

This investigation identified the types of drugs with the greatest presence in wastewater—analgesics and antibiotics, followed by antiepileptics and hormones with lower detection frequencies—highlighting the high concentrations of analgesics such as naproxen and paracetamol in municipal wastewater, as well as the high concentrations of antibiotics such as norfloxacin and ofloxacin in hospital wastewater. The most efficient methods for the elimination of drugs were physical ones such as the reverse osmosis and nanofiltration techniques, which reached removal rates ranging from 96 to 99.9% and from 85 to 99%, respectively; similarly, within the chemical methods, the best technique was ozonation, which achieved removal rates in the range of 90 to 99% for the proposed drugs, compared to biological methods, where the activated sludge technique showed highly variable removal rates depending on the characteristics of each drug (from 50 to 99 %), although it can be considered a good alternative to eliminate biodegradable drugs. The most efficient technique to eliminate drugs in municipal wastewater was reverse osmosis, achieving removal rates from 96% to 99.9% for ofloxacin, sulfamethoxazole, carbamazepine, and diclofenac; similarly, the most efficient technique for the removal of a wide variety of these drugs from hospital wastewater was activated sludge, since it eliminated analgesics and antibiotics in the range of 41–99%, but with lower efficiencies for antiepileptics and hormones. For future work, it is recommended to conduct a review of hybrid techniques to remove drugs from wastewater, since they could make up for the deficiencies of the individual techniques. To carry out a more in-depth analysis of drug removal techniques for municipal and hospital wastewaters, it would be convenient to search for articles in other databases and analyze closed-access studies.

## Figures and Tables

**Figure 1 ijerph-19-13105-f001:**
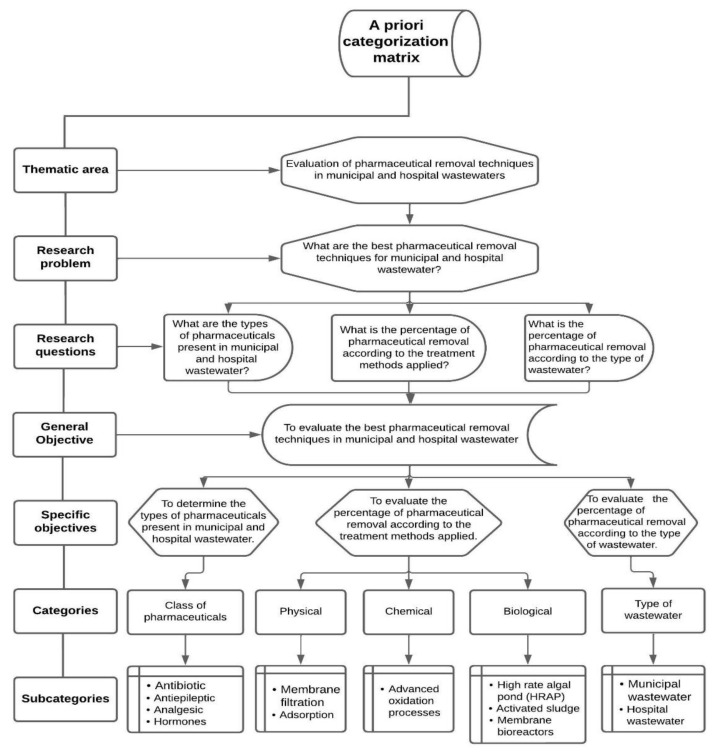
A priori categorization matrix.

**Figure 2 ijerph-19-13105-f002:**
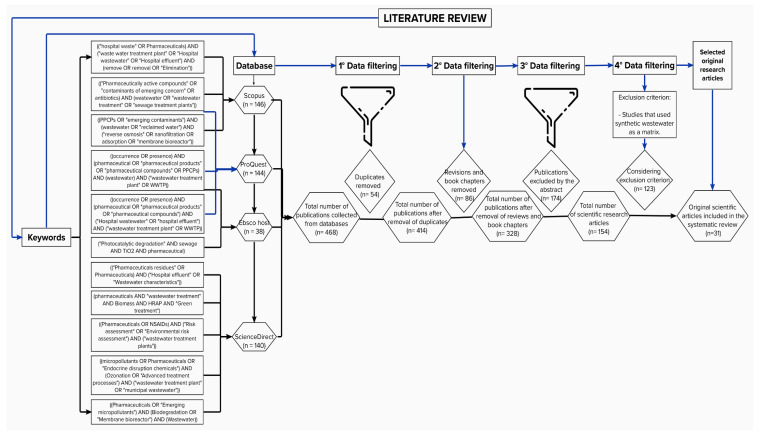
Outline of the search procedure and identification of articles included in this review.

**Figure 3 ijerph-19-13105-f003:**
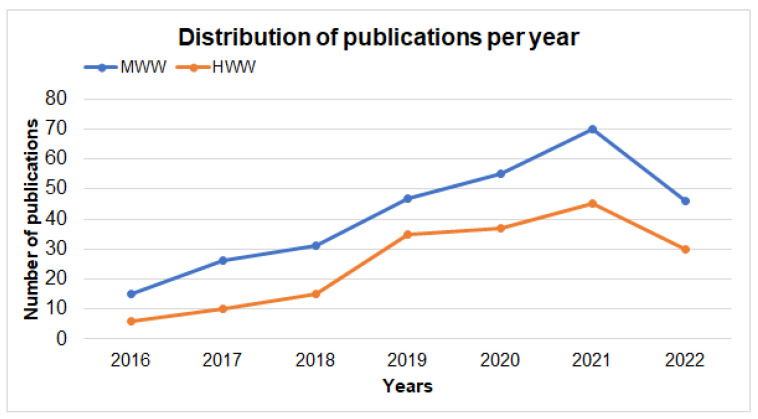
Publications related to techniques for the removal of pharmaceuticals from MWW and HWW. Source: own elaboration.

**Figure 4 ijerph-19-13105-f004:**
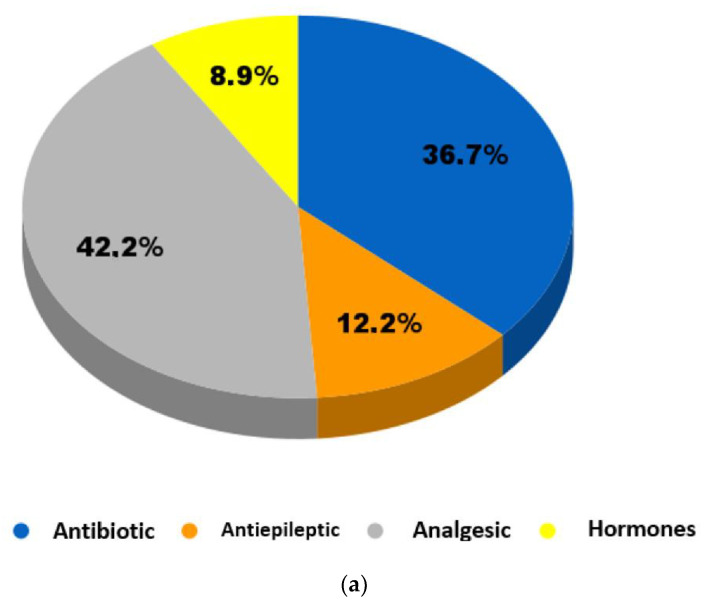
(**a**) Main therapeutic categories and (**b**) types of pharmaceuticals most frequently detected in wastewater.

**Figure 5 ijerph-19-13105-f005:**
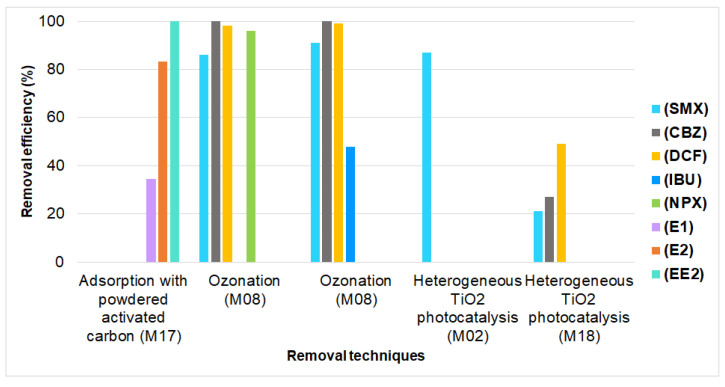
Removal of pharmaceuticals from municipal wastewater in studies with a pure experimental design (M17 = [49], M08 = [34], M02 = [28], M18 = [50]).

**Figure 6 ijerph-19-13105-f006:**
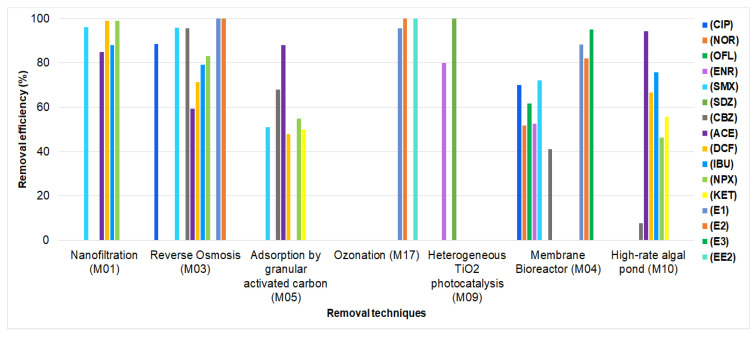
Removal of pharmaceuticals from municipal wastewater in studies with a quasi-experimental design (M01 = [27], M03 = [29], M05 = [31], M17 = [49], M09 = [35], M04 = [30], M10 = [36]).

**Figure 7 ijerph-19-13105-f007:**
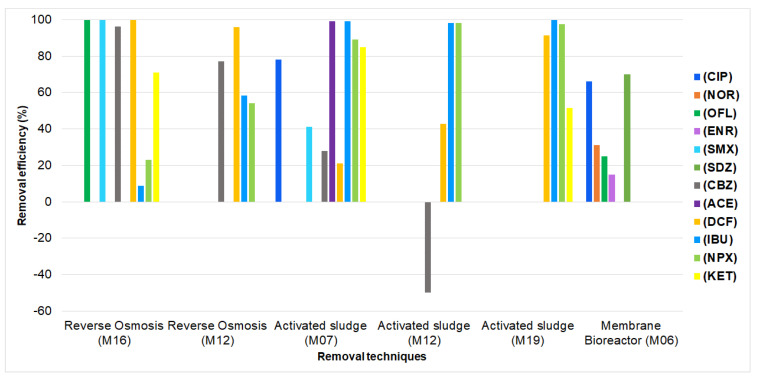
Removal of pharmaceuticals from municipal wastewater in studies with a pre-experimental design (M16 = [48], M12 = [41], M07 = [33], M12 = [41], M19 = [51], M06 = [32]).

**Table 1 ijerph-19-13105-t001:** Data collection form.

Code	Research Design	Type of Wastewater	Removal Method	Removal Technique	Type of Pharmaceutical (Range of Initial Concentration in ng/L) and (Range of Removal Percentage (%))																Ref.
					Antibiotic						Antiepileptic	Analgesic					Hormones				
					Ciprofloxacin (CIP)	Norfloxacin (NOR)	Ofloxacin (OFL)	Enrofloxacin (ENR)	Sulfamethoxazole (SMX)	Sulfadiazine (SDZ)	Carbamazepine (CBZ)	Paracetamol or Acetaminophen (ACE)	Diclofenac (DCF)	Ibuprofen (IBU)	Naproxen (NPX)	Ketoprofen (KET)	Estrone (E1)	17-β-estradiol (E2)	Estriol (E3)	17α-ethinylestradiol (EE2)	
H01	Pre-experimental	Hospital	Biological	Activated sludge					(<LOQ/2369); (−32%/77%)	(n.d./565.3); (−22/99%)	(41.1/720.7); (−116%/−9%)		(n.d./<LOQ); (-)								[23]
H02	Pre-experimental	Hospital	Biological	Activated sludge	(120/2000); (49%)	(100/1530); (71%)	(n.d./n.d.); (-)					(21,270/119,500); (88%)		(330/53,400); (68%)		(<LOQ/650); (64%)	(13/53); (28%)	(8/35); (35%)	(134/1480); (32%)	(2654/9833); (76%)	[24]
	Pre-experimental	Hospital	Biological	Activated sludge	(120/9110); (84%)	(70/820); (41%)	(n.d./n.d.); (-)					(11,060/57,650); (93%)		(1090/63,370); (97%)		(<LOQ/700); (55%)	(4/40); (83%)	(1/47); (-)	(27/512); (77%)	(881/1041); (26%)	[24]
H03	Quasi-experimental	Hospital	Chemical	Ozonation					(960); (90%)		(80); (90%)			(<LOQ); (-)							[25]
	Pure experimental	Hospital	Chemical	Ozonation					(750); (90%)		(30); (90%)			(850); (90%)							[25]
H04	Pre-experimental	Hospital	Biological	Activated sludge	(5600); (99%)				(30); (96%)		(151); (73%)	(12,400); (99%)									[26]
	Pre-experimental	Hospital	Biological	Activated sludge	(2180); (99%)				(132); (99%)		(73); (99%)	(12,300); (98%)									[26]
M01	Quasi-experimental	Municipal	Physical	Nanofiltration using TFC-SR2					(1000); (48%)			(1000); (38%)	(300); (58%)	(1000); (48%)	(300); (55%)						[27]
	Quasi-experimental	Municipal	Physical	Nanofiltration using NF-270					(1000); (95%)			(1000); (78%)	(300); (97%)	(1000); (84%)	(300); (96%)						[27]
	Quasi-experimental	Municipal	Physical	Nanofiltration using MPS-34					(1000); (96%)			(1000); (85%)	(300); (99%)	(1000); (88%)	(300); (99%)						[27]
M02	Pure experimental	Municipal	Chemical	Heterogeneous TiO_2_ photocatalysis					(100,000); (87%)												[28]
M03	Quasi-experimental	Municipal	Physical	Reverse osmosis	(570); (88.6%)				(920); (95.9%)		(200); (95.7%)	(27,000); (59.3%)	(160); (71.3%)	(12,000); (79.2%)	(8900); (83.1%)		(160); (100%)	(51); (100%)			[29]
M04	Quasi-experimental	Municipal	Biological	Membrane bioreactor (MBR)	(n.d./89); (70.1%)	(14/226); (51.8%)	(100/912); (61.8%)	(n.d./8); (52.7%)	(12/92); (72%)		(n.d./32); (41%)						(78/158); (88.2%)	(11/54); (82%)	(42/162); (95%)		[30]
M05	Quasi-experimental	Municipal	Physical	Adsorption by granular activated carbon (GAC)					(200,000); (51%)		(200,000); (68%)	(200,000); (88%)	(200,000); (48%)		(200,000); (55%)	(200,000); (50%)					[31]
M06	Pre-experimental	Municipal	Biological	Membrane Bioreactor (MBR)	(220); (66%)	(620); (31%)	(1020); (25%)	(27); (15%)		(410); (70%)											[32]
M07	Pre-experimental	Municipal	Biological	Activated sludge	(n.d./4200); (78%)				(n.d./5300); (41%)		(820/6500); (28%)	(55,000/623,000); (99%)	(460/6500); (21%)	(8000/53,000); (99%)	(n.d./38,000); (89%)	(n.d./1700); (85%)					[33]
M08	Pure experimental	Municipal	Chemical	Ozonation					(263); (86%)		(894); (100%)		(1138); (98%)	(<LOQ); (-)	(772); (96%)						[34]
	Pure experimental	Municipal	Chemical	Ozonation					(263); (91%)		(894); (100%)		(1138); (99%)	(878); (48%)	(<LOQ); (-)						[34]
M09	Quasi-experimental	Municipal	Chemical	Heterogeneous TiO_2_ photocatalysis				(1,000,000); (80%)		(1,000,000); (100%)											[35]
M10	Quasi-experimental	Municipal	Biological	High-rate algal pond (HRAP)							(100/350); (7.7)	(210/120,000); (94.4)	(130/470); (66.6)	(1400/4600); (75.7)	(3200/8100); (46.3)	(110/120); (55.8)					[36]
H05	Pre-experimental	Hospital							(20,590); (-)	(3440); (-)											[37]
	Pre-experimental	Municipal							(7800); (-)	(70); (-)											[37]
H06	Pre-experimental	Hospital	Biological	Activated sludge	(n.d./120,000); (-)	(340,000/526,000); (-)	(124,000/198,000); (-)														[38]
	Pre-experimental	Hospital	Biological	Membrane bioreactor (MBR)	(n.d./100,000); (-)	(372,000/561,000); (-)	(n.d./147,000); (-)														[38]
M11	Quasi-experimental	Municipal	Physical	Nanofiltration	(135/3150); (99%)																[39]
H07	Pre-experimental	Municipal	Biological	Activated sludge	(346); (-)						(348); (-)										[40]
	Pre-experimental	Hospital			(2550); (-)						(880); (-)										[40]
	Pre-experimental	Hospital			(5360); (-)						(45); (-)										[40]
M12	Pre-experimental	Municipal	Biological	Activated sludge							(45/2394); (−50%)		(6/708); (42.9%)	(12/147,500); (98%)	(4910/ 521,700); (98%)						[41]
	Pre-experimental	Municipal	Physical	Reverse osmosis							(78/4687); (77%)		(34/288); (96%)	(9/1184); (58.3%)	(57/1311); (54.1%)						[41]
H08	Pre-experimental	Municipal	Biological	Activated sludge									(48,000); (-)	(45,000); (-)							[42]
	Pre-experimental	Hospital	Biological	Membrane bioreactor (MBR)									(39,000); (-)	(36,000); (-)							[42]
	Pre-experimental	Hospital	Biological	Activated sludge									(166,000); (-)	(32,000); (-)							[42]
H09	Pure experimental	Hospital	Chemical	Heterogeneous TiO_2_ photocatalysis	(3,000,000); (86.6%)																[43]
H10	Pre-experimental	Hospital										(2660); (-)	(590); (-)	(620); (-)	(1790); (-)			(80); (-)			[44]
M13	Quasi-experimental	Municipal	Biological	High-rate algal pond (HRAP)					(69.4/902.5); (50.5%)		(9.6/42.4); (0%)	(156.4/7781.1); (100%)	(270.8/2117.8); (54.8%)	(713.3/23,811.4); (79%)							[45]
M14	Quasi-experimental	Municipal	Biological	High-rate algal pond (HRAP)	(332/710); (68%)						(502/557); (−14%)	(12,133/15,611); (100%)	(732/1030); (46%)								[46]
M15	Pre-experimental	Municipal	Biological	Activated sludge									(n.d./410,000); (-)	(50/94,000); (-)	(300/1,370,000); (-)	(50/260,000); (-)					[47]
	Pre-experimental	Municipal	Biological	Activated sludge									(n.d./82,000); (-)	(n.d./23,000); (-)	(n.d./67,000); (-)	(40/37,000); (-)					[47]
	Pre-experimental	Municipal	Biological	Activated sludge									(n.d./45,000); (-)	(50/13,000); (-)	(60/26,000); (-)	(10/10,500); (-)					[47]
M16	Pre-experimental	Municipal	Physical	Reverse osmosis	(<LOQ); (-)		(134); (99.9%)		(150); (99.9%)		(412); (96.1%)		(86); (99.9%)	(78); (8.9%)	(144); (22.9%)	(616); (71.1%)					[48]
M17	Quasi-experimental	Municipal	Chemical	Ozonation													(3.95); (95.7%)	(4.68); (99.9%)		(0.24); (99.9%)	[49]
	Pure experimental	Municipal	Physical	Adsorption with powdered activated carbon (PAC)													(3.95); (34.4%)	(4.68); (83.3%)		(0.24); (99.9%)	[49]
M18	Pure experimental	Municipal	Chemical	Heterogeneous TiO_2_ photocatalysis					(190,000); (21%)		(198,000); (27%)		(194,000); (49 %)								[50]
M19	Pre-experimental	Municipal	Biological	Activated sludge							(<LOQ); (0%)		(77.3); (91.4%)	(434); (99.9%)	(444); (97.4%)	(39.1); (51.4%)					[51]
M20	Pre-experimental	Municipal			(157/320); (-)		(213/677); (-)		(44.6/116); (-)		(57.8/133); (-)	(20,710/67,340); (-)	(27/44); (-)	(n.d./4750); (-)	(1805/4090); (-)	(77.6/150.6); (-)					[52]
	Pre-experimental	Municipal			(<LOQ/270); (-)		(75/360); (-)		(76/369); (-)		(110/220); (-)	(33,180/53,000); (-)	(14/53); (-)	(75/5990); (-)	(2000/6085); (-)	(214/461); (-)					[52]
	Pre-experimental	Municipal			(27/62); (-)		(58/205); (-)		(99/150); (-)		(99/236); (-)	(15,990/54,680); (-)	(16/68); (-)	(<LOQ/2580); (-)	(1490/3110); (-)	(149/363); (-)					[52]
M21	Quasi-experimental	Municipal	Biological	Membrane bioreactor (MBR)							(630); (−25%)		(1840); (38%)								[53]

**Table 2 ijerph-19-13105-t002:** Concentration ranges of pharmaceuticals present in the influent of municipal WWTPs.

Class of Pharmaceuticals	Pharmaceutical Drug	Concentration Range (ng/L)	Reference
Antibiotic	CIP	27–4200	[30,32,33,40,46,52]
	NOR	14–620	[30,32]
	OFL	58–1020	[30,32,52]
	ENR	8–27	[30,32]
	SMX	12–7800	[30,33,37,45,52]
	SDZ	70–410	[32,37]
Antiepileptic	CBZ	96–6500	[30,33,40,41,45,46,51,52]
Analgesic	ACE	156.4–623,000	[33,45,46,52]
	DCF	6–410000	[33,41,42,45,46,47,51,52]
	IBU	50–147,500	[33,41,42,45,47,51,52]
	NPX	60–1,370,000	[33,41,47,51,52]
	KET	10–260,000	[33,47,51,52]
Hormones	E1	78–158	[30]
	E2	11–54	[30]
	E3	42–162	[30]

**Table 3 ijerph-19-13105-t003:** Concentration ranges of pharmaceuticals present in the influent of hospital WWTPs.

Class of Pharmaceuticals	Pharmaceutical Drug	Concentration Range (ng/L)	Reference
Antibiotic	CIP	120–120,000	[24,26,38,40]
	NOR	70–561,000	[24,38]
	OFL	124000–198,000	[24,38]
	SMX	30–20590	[23,26,37]
	SDZ	565.3–3440	[23,37]
Antiepileptic	CBZ	41.1–880	[23,26,40]
Analgesic	ACE	2660–119,500	[24,26,44]
	DCF	590–166,000	[23,42,44]
	IBU	330–63370	[24,42,44]
	KET	650–700	[24]
Hormones	E1	4–53	[24]
	E2	1–80	[24,44]
	E3	27–1480	[24]
	EE2	881–9833	[24]

**Table 4 ijerph-19-13105-t004:** Removal of pharmaceuticals by physical treatment methods.

Research Design	Technique	Class of Pharmaceuticals	Pharmaceutical Drug	Type of Wastewater	Initial Concentration (ng/L)	Removal Efficiency (%)	Reference
Pre–experimental	Reverse osmosis	Antibiotic	OFL	Municipal	134	99.9	[48]
			SMX		150	99.9	[48]
		Antiepileptic	CBZ		78/4687	77	[41]
					412	96.1	[48]
		Analgesic	DCF		34/288	96	[41]
					86	99.9	[48]
			IBU		9/1184	58.3	[41]
					78	8.9	[48]
			NPX		57/1311	54.1	[41]
					144	22.9	[48]
			KET		616	71.1	[48]
Quasi-experimental	Reverse osmosis	Antibiotic	CIP	Municipal	570	88.6	[29]
			SMX		920	95.9	[29]
		Antiepileptic	CBZ		200	95.7	[29]
		Analgesic	ACE		27,000	59.3	[29]
			DCF		160	71.3	[29]
			IBU		12,000	79.2	[29]
			NPX		8900	83.1	[29]
		Hormones	E1		160	100	[29]
			E2		51	100	[29]
	Nanofiltration	Antibiotic	CIP	Municipal	135 /3150	99	[39]
			SMX		1000	48	[27]
					1000	95	[27]
					1000	96	[27]
		Analgesic	ACE		1000	38	[27]
					1000	78	[27]
					1000	85	[27]
			DCF		300	58	[27]
					300	97	[27]
					300	99	[27]
			IBU		1000	48	[27]
					1000	84	[27]
					1000	88	[27]
			NPX		300	55	[27]
					300	96	[27]
					300	99	[27]
	Adsorption by granular activated carbon (GAC)	Antibiotic	SMX	Municipal	200,000	51	[31]
		Antiepileptic	CBZ		200,000	68	[31]
		Analgesic	ACE		200,000	88	[31]
			DCF		200,000	48	[31]
			NPX		200,000	55	[31]
			KET		200,000	50	[31]
Pure experimental	Adsorption with powdered activated carbon (PAC)	Hormones	E1	Municipal	3.95	34.4	[49]
			E2		4.68	83.3	[49]
			EE2		0.24	99.9	[49]

**Table 5 ijerph-19-13105-t005:** Removal of pharmaceuticals by chemical treatment methods.

Research Design	Technique	Class Of Pharmaceuticals	Pharmaceutical Drug	Type Of Wastewater	Initial Concentration (ng/L)	Removal Efficiency (%)	Reference
Quasi-experimental	Ozonation	Antibiotic	SMX	Hospital	960	90	[25]
		Antiepileptic	CBZ		80	90	[25]
		Hormones	E1	Municipal	3.95	95.7	[49]
			E2		4.68	99.9	[49]
			EE2		0.24	99.9	[49]
Pure experimental	Heterogeneous TiO_2_ photocatalysis	Antibiotic	ENR	Municipal	1,000,000	80	[35]
			SDZ		1,000,000	100	[35]
	Ozonation	Antibiotic	SMX	Hospital	750	90	[25]
				Municipal	263	86	[34]
					263	91	[34]
		Antiepileptic	CBZ	Hospital	30	90	[25]
				Municipal	894	100	[34]
					894	100	[34]
		Analgesic	DCF	Municipal	1138	98	[34]
					1138	99	[34]
			IBU	Hospital	850	90	[25]
					878	48	[34]
			NPX	Municipal	772	96	[34]
	Heterogeneous TiO_2_ photocatalysis	Antibiotic	CIP	Hospital	3,000,000	86.6	[43]
			SMX	Municipal	100,000	87	[28]
					190,000	21	[50]
		Antiepileptic	CBZ	Municipal	198,000	27	[50]
		Analgesic	DCF	Municipal	194,000	49	[50]

**Table 6 ijerph-19-13105-t006:** Removal of pharmaceuticals by biological treatment methods.

Research Design	Technique	Class of Pharmaceuticals	Pharmaceutical Drug	Type of Wastewater	Initial Concentration (ng/L)	Removal Efficiency (%)	Reference
Pre-experimental	Activated sludge	Antibiotic	CIP	Hospital	120/2000	49	[24]
					120/9110	84	[24]
					5600	99	[26]
					2180	99	[26]
				Municipal	n.d./4200	78	[33]
			NOR	Hospital	100/1530	71	[24]
					70/820	41	[24]
			SMX	Hospital	<LOQ/2369	−32/77	[23]
					30	96	[26]
					132	99	[26]
				Municipal	n.d./5300	41	[33]
			SDZ	Hospital	n.d./565.3	−22/99	[23]
		Antiepileptic	CBZ	Hospital	41.1/720.7	−116/-9	[23]
					151	73	[26]
					73	99	[26]
				Municipal	820/6500	28	[33]
					45/2394	-50	[41]
		Analgesic	ACE	Hospital	21,270/119,500	88	[24]
					11,060/57,650	93	[24]
					12400	99	[26]
					12300	98	[26]
				Municipal	55,000/623,000	99	[33]
			DCF	Municipal	460/6500	21	[33]
					6/708	42.9	[41]
					77.3	91.4	[51]
			IBU	Hospital	330/53,400	68	[24]
					1090/63370	97	[24]
				Municipal	8000/53,000	99	[33]
					128/147,500	98	[41]
					434	99.9	[51]
			NPX	Municipal	n.d./38000	89	[33]
					4910/ 521,700	98	[41]
					444	97.4	[51]
			KET	Hospital	<LOQ/650	64	[24]
					<LOQ/700	55	[24]
				Municipal	n.d./1700	85	[33]
					39.1	51.4	[51]
		Hormones	E1	Hospital	13/53	28	[24]
					4/40	83	[24]
			E2		8/35	35	[24]
			E3		134/1480	32	[24]
					27/512	77	[24]
			EE2		2654/9833	76	[24]
					881/1041	26	[24]
	Membrane bioreactor (MBR)	Antibiotic	CIP	Municipal	220	66	[32]
			NOR		620	31	[32]
			OFL		1020	25	[32]
			ENR		27	15	[32]
			SDZ		410	70	[32]
Quasi-experimental	Membrane bioreactor (MBR)	Antibiotic	CIP	Municipal	n.d./89	70.1	[30]
			NOR		14/226	51.8	[30]
			OFL		100/912	61.8	[30]
			ENR		n.d./89	52.7	[30]
			SMX		12/92	72	[30]
		Antiepileptic	CBZ		n.d./32	41	[30]
					630	−25	[53]
		Analgesic	DCF		1840	38	[53]
		Hormones	E1		78/158	88.2	[30]
			E2		11/54	82	[30]
			E3		42/162	95	[30]
	High-rate algal pond (HRAP)	Antibiotic	CIP	Municipal	332/710	68	[46]
			SMX		69.4/902.5	50.5	[45]
		Antiepileptic	CBZ		100/350	7.7	[36]
					9.6/42.4	0	[45]
					502/557	−14	[46]
		Analgesic	ACE		210/120,000	94.4	[36]
					156.4/7781.1	100	[45]
					12,133/15,611	100	[46]
			DCF		130/470	66.6	[36]
					270.8/2117.8	54.8	[45]
					732/1030	46	[46]
			IBU		1400/4600	75.7	[36]
					713.3/23811.4	79	[45]
			NPX		3200/8100	46.3	[36]
			KET		110/120	55.8	[36]

n.d.: Undetected. <LOQ: below the limit of quantification.

**Table 7 ijerph-19-13105-t007:** Removal of pharmaceuticals from hospital wastewater.

Research Design	Techniques	Class of Pharmaceuticals	Pharmaceutical Drug	Removal Efficiency (%)	Reference
Pre-experimental	Activated sludge	Antibiotic	CIP	49	[24]
				84	[24]
				99	[26]
				99	[26]
			NOR	71	[24]
				41	[24]
			SMX	−32/77	[23]
				96	[26]
				99	[26]
			SDZ	−22/99	[23]
		Antiepileptic	CBZ	−116/−9	[23]
				73	[26]
				99	[26]
		Analgesic	ACE	88	[24]
				93	[24]
				99	[26]
				98	[26]
			IBU	68	[24]
				97	[24]
			KET	64	[24]
				55	[24]
		Hormones	E1	28	[24]
				83	[24]
			E2	35	[24]
			E3	32	[24]
				77	[24]
			EE2	76	[24]
				26	[24]
Quasi-experimental	Ozonation	Antibiotic	SMX	90	[25]
		Antiepileptic	CBZ	90	[25]
Pure experimental	Ozonation	Antibiotic	SMX	90	[25]
		Antiepileptic	CBZ	90	[25]
		Analgesic	IBU	90	[25]
	Heterogeneous TiO_2_ photocatalysis	Antibiotic	CIP	86.6	[43]

## Data Availability

Not applicable.

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
