# Peer review of "Literature Review: Evaluation of Drug Removal Techniques in Municipal and Hospital Wastewater"

_ijerph, 2022, doi:10.3390/ijerph192013105_

Round 1

Reviewer 1 Report

The manuscript describes a review of literature about different waste-water treatment systems and focuses the discussion on the pharmacological compounds' remotion efficiency. In general, the manuscript is pertinent and provides interesting information that allows us to see how the distinct water treatment strategies are used and which could be the advantages or disadvantages of the remotion of drugs. However, the manuscript presents several structural odds to be accepted, at least in the current version.

Why did the authors not consider including the use of white-rot fungi or enzymatic treatments in their review???? These are relevant biological strategies for xenobiotics pollutants remotion, including pharmacologic compounds; broad scientific literature exists on these themes. I suggest including this important theme to complete the review, or at least add the argument explaining why the authors did not consider this in the revision.

L24: Please, describe the BOD abbreviation the first time you mention it in the text.

L39: What does “EPs” mean?

L42: please change “pharmaceuticals” to “drugs”

Table 1: I suggest putting this information represented in a scheme or flux diagram to improve its visualization.

Table 2: I suggest eliminating this table: the information is better explained in the Figure 1 scheme. Why did the authors not use the same keywords combinations for all databases? Please explain this.

Table 3 could be added as supplementary material.

Table 4 is absolutely unreadable; please correct format.

Correct the under-script number for the chemical formulas (TiO2).

Figure 1 is a good scheme, but the authors must improve the quality of the figure.

Homolog use of points after the letters in abbreviatures; sometimes are without points, and there are written with points.

Authors must describe what they consider as a “pre-experimental design”, “quasi-experimental design” or “pure experimental design”.

L186-190: Authors must be most careful about arguments to explain the results observed, especially when they are unexpected, such as the negative remotion rate reported. They could contrast the results with other reports trying to explain the reason for the augmentation of drug concentration after treatment; after all, this review objective is to elucidate the most efficient water treatment systems to remove drugs to avoid any environmental impact when effluents are discharged into superficial water bodies. Also, authors must cite appropriate papers to underlie their arguments; in this case, the article cited (Wang et al. 2018) on one side does not approach any drug remotion analysis, and on the other, never argue any like “may be because the metabolites of these pharmaceuticals are transformed back to the parent compound due to microbial activity”, which is in this case, highly debatable.

Authors must describe which information was considered to include in the analysis for performing the graphics (Figs. 7, 8, and 9). Was there considered the result from a paper for each drug? Or were there considered the mean of different results from papers reporting the same drug? Authors must mention this in the methods section. In this case, it is desirable to include more than one data for each compound; authors must add standard deviations (SD) bars for each drug.

Figure 7: why are there two groups of “Ozonation (M08)” and two of “Heterogeneous TiO2…”??? is there any reason for that? Which is the difference? Authors must be clear on this.

Fig. 9: Which is the difference between activated sludge M07, M12, and M19???

Author Response

Dear colleague, it cost a lot of work to make the suggested changes; but the group of researchers responded to all the comments made, which are:

Why did the authors not consider including the use of white-rot fungi or enzymatic treatments in their review???? These are relevant biological strategies for xenobiotics pollutants remotion, including pharmacologic compounds; broad scientific literature exists on these themes. I suggest including this important theme to complete the review, or at least add the argument explaining why the authors did not consider this in the revision.

Ans. It has not been considered to include the use of white rot fungi or enzymatic treatments because in the investigation only studies that had real wastewater as matrix were taken, in this sense, although there have been advances in the work focused on the bioremediation of drugs using mushrooms in recent years, there are few reports using non-sterile conditions and real wastewater. In this same context, Grandclement et al. (2017), point out that the performance of white rot fungi (WRF) for drug removal has often been studied in synthetic wastewater containing high concentrations of contaminants (up to several mg/L) under sterile conditions to avoid contamination. contamination with bacterial strains

L24: Please, describe the BOD abbreviation the first time you mention it in the text.

Ans. Due to the suggestion of another reviewer the introduction was corrected.

L39: What does “EPs” mean?

Ans. Due to the suggestion of another reviewer the introduction was corrected.

L42: please change “pharmaceuticals” to “drugs”

Ans. Ok

Table 1: I suggest putting this information represented in a scheme or flux diagram to improve its visualization.

Ans. The authors agree with the suggestion provided and the changes were made.

Table 2: I suggest eliminating this table: the information is better explained in the Figure 1 scheme. Why did the authors not use the same keywords combinations for all databases? Please explain this.

Ans. The authors agree with their point of view and table N 02 was changed to a scheme for better understanding; Regarding the keywords, the same ones were not used in all the databases because the information necessary for the development of the objectives demanded the use of specific terms related to the different techniques.

Table 3 could be added as supplementary material.

Ans. The authors agree that Table 3 is included in supplementary material.

Table 4 is absolutely unreadable; please correct format.

Ans. Changes were made

Correct the under-script number for the chemical formulas (TiO2).

Ans. Changes were made

Figure 1 is a good scheme, but the authors must improve the quality of the figure.

Ans. Changes were made

Homolog use of points after the letters in abbreviatures; sometimes are without points, and there are written with points.

Ans. Changes were made

Authors must describe what they consider as a “pre-experimental design”, “quasi-experimental design” or “pure experimental design”.

Ans. To place the research design in the tables and figures, the authors were guided by the existing information in the same documents investigated.
• Pre-experimental design: In this type of research, no variable is controlled, they are practically full-scale research.
• Quasi-experimental design: In this type of research some variables are controlled.
• Pure experimental design: In this type of research all the variables are controlled, they are studies carried out in a laboratory.

L186-190: Authors must be most careful about arguments to explain the results observed, especially when they are unexpected, such as the negative remotion rate reported. They could contrast the results with other reports trying to explain the reason for the augmentation of drug concentration after treatment; after all, this review objective is to elucidate the most efficient water treatment systems to remove drugs to avoid any environmental impact when effluents are discharged into superficial water bodies. Also, authors must cite appropriate papers to underlie their arguments; in this case, the article cited (Wang et al. 2018) on one side does not approach any drug remotion analysis, and on the other, never argue any like “may be because the metabolites of these pharmaceuticals are transformed back to the parent compound due to microbial activity”, which is in this case, highly debatable.

Ans. The changes were made, leaving the support paragraph worded as follows:

L 223-131: It should be noted that the biological treatment techniques showed negative removals for drugs such as SMX, SDZ and CBZ (Table 6), which indicates that these micropollutants had higher concentrations in the effluents than in the influents, in relation to it. , Blair et al. (2015) argued that among the reasons for negative removal is the fact that drugs may be encased in fecal particles and when microbes break down this feces, the drugs are released. Another reason for this phenomenon is that the drug metabolites would be transformed back into the original compounds by the action of microorganisms (Verlicchi et al., 2012). In addition to this, for Clara et al. (2004) negative removals are also attributed to daily fluctuations in concentration during the sampling period.

Authors must describe which information was considered to include in the analysis for performing the graphics (Figs. 7, 8, and 9). Was there considered the result from a paper for each drug? Or were there considered the mean of different results from papers reporting the same drug? Authors must mention this in the methods section. In this case, it is desirable to include more than one data for each compound; authors must add standard deviations (SD) bars for each drug.

Ans. The numbering for the graphs was changed, now they are figures 5, 6 and 7. To make these figures, the percentage of removal (single value) reported in each of the articles for the drugs under study was considered, additionally, it should be emphasized that each figure corresponds to studies with the same research design (Pure Experimental, Quasi-experimental, Pre-experimental).
On the other hand, the information was taken as it is in the articles, which reported a single removal value for each drug, so it is not necessary to make averages or standard deviation bars.

Figure 7: why are there two groups of “Ozonation (M08)” and two of “Heterogeneous TiO2…”??? is there any reason for that? Which is the difference? Authors must be clear on this.

Ans. The reason for there being two ozonation groups of M08 is that within one of the consulted articles, two experiments were carried out, considering two different samples of municipal wastewater, in addition, different doses of Ozone were taken into account for each experiment. In relation to the Heterogeneous photocatalysis technique with TiO2, there are two groups because the results of two different investigations are represented.

Fig. 9: Which is the difference between activated sludge M07, M12, and M19???

Ans. The difference is that each one represents an article with different authors and different results.

best regards

Reviewer 2 Report

The topic of the paper is interesting and suitable for the journal. The authors make a literature review related to drug removal techniques (including physical, chemical and biological) from wastewater, considering both municipal wastewater and hospital wastewater. 

After careful reading the submitted paper I would recommend major revision.

Main issues that should be addressed:

(1)    The abstract and keywords do NOT correspond to the submitted paper. They are related to biosensors for monitoring wastewater pollutants, and this topic has nothing to do with the title and content of the submitted paper. Please provide the correct abstract and keywords

(2)    Table 4 is completely unconfigured and, thus, it is not easy to read or follow the information provided. Please, configure Table 4 so all the information could be read and understood. Maybe you need to split the table into two parts, or use a page of higher size (A3 instead of A4).

(3)    The numbering of the lines and pages starts again after Table 4. Please solve this issue.

(4)    Line 9 (after Table 4) “applying the established exclusion criterion”…. Please, clearly indicate which is this exclusion criterion.

(5)    Line 16 (after Table 4) “…can be seen in the thematic maps produced”. Where are these maps?

(6)    Line 24 (after Table 4)… shows an exponential increase in recent years (Figure 2). According to Figure 2, the increase is lineal not exponential. Please check the axis of this figure or change the text to reflect what can be seen in the Figure.  

(7)    Clearly define these terms:

a.       Pre-experimental

b.       Quasi-experimental

c.       Pure experimental

(8)    It would be useful to make several distinctions among the references, the scale of the research (lab-scale/pilot-plant scale/demonstration plant/full-scale plant) and the type of wastewater (real wastewater/synthetic wastewater/mixed wastewater (real plus artificially modified).

(9)    In figures 7, 8 and 9, there should be some variability in the removal for a given pharmaceutical using a given removal technique. However, the bars do not include this variability which is important. E.g. The removal of DCF by activated sludge is not going to be always 21%.... there must be some differences among the studies that report the same pollutant removal with the same technology. Please reflect this variability in the Figures.

(10)The major advantages, disadvantages, and cost of each pharmaceutical removal technology should be included, discussed, and probably summarized in a Table. Being the main aim of the paper to show the best techniques for removing pharmaceuticals from municipal and hospital facilities… if the costs are neither considered not shown, we really do not know if the removal technique is feasible or not.

(11) Check all the references…  e.g. in line 46 -Soares et al [8] in the reference section, the number [8] correspond to Silva et al. In line 51 Krako et al [the reference number is missing]. Line 25… with Alvarino et al [the reference number is also missing]…

Author Response

Dear colleague, I send the answers to each of the comments made

(1)    The abstract and keywords do NOT correspond to the submitted paper. They are related to biosensors for monitoring wastewater pollutants, and this topic has nothing to do with the title and content of the submitted paper. Please provide the correct abstract and keywords

Ans. Suggested changes are made

(2)    Table 4 is completely unconfigured and, thus, it is not easy to read or follow the information provided. Please, configure Table 4 so all the information could be read and understood. Maybe you need to split the table into two parts, or use a page of higher size (A3 instead of A4).

  Ans. The table was modified.

(3)    The numbering of the lines and pages starts again after Table 4. Please solve this issue.

Ans. Suggested changes are made

(4)    Line 9 (after Table 4) “applying the established exclusion criterion”…. Please, clearly indicate which is this exclusion criterion.

  Ans. The researchers considered including the exclusion criteria in the methodological part.
Which are worded as follows:
L108-110: “Furthermore, the use of synthetic wastewater as a matrix and the use of hybrid techniques for the elimination of drugs in the investigations were considered as exclusion criteria for this systematic review.

(5)    Line 16 (after Table 4) “…can be seen in the thematic maps produced”. Where are these maps?

Ans. Removed text referring to these thematic maps.

(6)    Line 24 (after Table 4)… shows an exponential increase in recent years (Figure 2). According to Figure 2, the increase is lineal not exponential. Please check the axis of this figure or change the text to reflect what can be seen in the Figure.  

  Ans. The text was changed.

(7)    Clearly define these terms:

  1. Pre-experimental
  2. Quasi-experimental
  3. Pure experimental

Ans.It is defined in the papers used to write the manuscript.

• Pre-experimental design: In this type of research, no variable is controlled, they are practically full-scale research.
• Quasi-experimental design: In this type of research some variables are controlled.
• Pure experimental design: In this type of research all the variables are controlled, they are studies carried out in a laboratory.

(8)    It would be useful to make several distinctions among the references, the scale of the research (lab-scale/pilot-plant scale/demonstration plant/full-scale plant) and the type of wastewater (real wastewater/synthetic wastewater/mixed wastewater (real plus artificially modified).

Ans. The distinction that was taken into account for this research was the design of the research, separating those investigations that were carried out in a laboratory (pure experimental design), in a pilot plant (quasi-experimental design) and full-scale plants (pre-experimental design). -experimental). Regarding the type of wastewater, all the investigations used real wastewater as a matrix.

(9)    In figures 7, 8 and 9, there should be some variability in the removal for a given pharmaceutical using a given removal technique. However, the bars do not include this variability which is important. E.g. The removal of DCF by activated sludge is not going to be always 21%.... there must be some differences among the studies that report the same pollutant removal with the same technology. Please reflect this variability in the Figures.

Ans. Now these figures are 5, 6 and 7, on the other hand, this variability cannot be reflected because the bars represent the removal values obtained by each of the authors of the investigations and do not represent the values of several authors who used the same technique. In addition, the authors reported a single percentage of removal for each drug in their investigations, these data were reflected as they are shown in the results of these articles.

(10)The major advantages, disadvantages, and cost of each pharmaceutical removal technology should be included, discussed, and probably summarized in a Table. Being the main aim of the paper to show the best techniques for removing pharmaceuticals from municipal and hospital facilities… if the costs are neither considered not shown, we really do not know if the removal technique is feasible or not.

Ans. For this review, it was not considered to evaluate the issue of the costs involved in each removal technique and the feasibility of the techniques was evaluated taking into account their application in real-scale investigations and the removal percentages achieved. Likewise, the suggestion will be taken into account for the realization of a future investigation.

(11) Check all the references…  e.g. in line 46 -Soares et al [8] in the reference section, the number [8] correspond to Silva et al. In line 51 Krako et al [the reference number is missing]. Line 25… with Alvarino et al [the reference number is also missing]…

Ans. References have been corrected.

Reviewer 3 Report

In this paper the authors a) perform a literature search for methods used to treat municipal and hospital wastewater, and b) attempt to compare different methods for their effectiveness. A number of potential improvements to this manuscript should be considered.
a) The Abstract repeatedly mention biosensors and their application, but the manuscript does not discuss this topic.
b) The English appears to be acceptable until sentences are read more carefully, in which case the meanings are often hard to interpret. One particular improvement would to be careful of introductory phrases, as frequently they don't seem to refer to anything whereas they should refer to particular topics in a preceeding discussion or sentence.
c) The literature search portion of the manuscript is discussed extensively (with tables). However, given that this is part of the data collection, a number of things are not covered adequately. i) Why are different search terms used for different databases? ii) More than half of the papers were excluded because of the Abstract content. I can understand that in a wide search there could be a sizeable number of papers that are "hits" which do not contain the expected information. But with over half the papers excluded this way, could there be a failure in choosing the best search parameters? This exclusion of data is not sufficiently justified in the manuscript. iii) Figure 1 deals with the number of hits over the years selected for this study, showing an increase of papers published as an indication of interest in this area. I would argue that papers (that were excluded by the authors' criteria) using artificial wastewater show interest in this area, as model studies typically come before practical applications. This means that using Figure 1 in this way could be misleading.
d) The summary of the studies on removal of pollutants represents a lot of work and seems to be well organized. However, conclusions drawn from these tables were often unclear, possibly due to problems in wording. i) PhCs were referred to as being more/most frequent, but it wasn't clear what that meant. Did these appear in the greatest proportion of publications? Were they in the greatest concentrations? ii) Different PhCs varied in concentrations from 2 to 6 orders of magnitude from study to study with no means or other type of analyses, so it is not clear to me how these numbers can be compared. iii) For Fig. 8, the claim was made that RO had the highest removal rate, but the way I viewed this graph, it appeared that ozonation and nanofiltration were as effective. This should be clarified. 
e) Since this paper seems to take the wide view, looking at both the frequency to which publications appeared and the efficacy of the methods used, there should be one more important point addressed which at best is skimmed over briefly. There is a difference between removal of pollutants and destruction of pollutants. Some methods merely concentrate the pollutants while separating them out from the waste stream; are these the "best" methods if the pollutants still need to be dealt with? Also, any chemical change means that the detection method no longer identifies them. That could be that even 100% "destruction" could still mean a very large amount of potentially bioactive breakdown products are still present. This issue should be mentioned when discussing effectiveness of methods.

Author Response

Dear colleague, the suggested changes were made:

a) The Abstract repeatedly mention biosensors and their application, but the manuscript does not discuss this topic.

Ans. this bug was fixed.
b) The English appears to be acceptable until sentences are read more carefully, in which case the meanings are often hard to interpret. One particular improvement would to be careful of introductory phrases, as frequently they don't seem to refer to anything whereas they should refer to particular topics in a preceeding discussion or sentence.

Ans. was improved in many paragraphs.
c) The literature search portion of the manuscript is discussed extensively (with tables). However, given that this is part of the data collection, a number of things are not covered adequately.

i) Why are different search terms used for different databases?

Ans. The same search terms were not used in all the databases because the information necessary for the development of the objectives required the use of specific terms related to the different techniques.

ii) More than half of the papers were excluded because of the Abstract content. I can understand that in a wide search there could be a sizeable number of papers that are "hits" which do not contain the expected information. But with over half the papers excluded this way, could there be a failure in choosing the best search parameters? This exclusion of data is not sufficiently justified in the manuscript.

Ans. The research team considered the exclusion of these articles because in their abstract they showed that they studied the removal of drugs using hybrid techniques, which combine one or more of the removal methods (physical, chemical and biological) and our objective was to evaluate each technique and its respective removal method separately. Likewise, articles were excluded that in the summary detailed that they used synthetic wastewater or another type of wastewater that was not municipal or hospital as a matrix.
In addition, the following paragraph was added to the manuscript:
L108-110: "Likewise, the use of synthetic wastewater as a matrix and the use of hybrid techniques for drug removal in research were considered as exclusion criteria for this systematic review."

iii) Figure 1 deals with the number of hits over the years selected for this study, showing an increase of papers published as an indication of interest in this area. I would argue that papers (that were excluded by the authors' criteria) using artificial wastewater show interest in this area, as model studies typically come before practical applications. This means that using Figure 1 in this way could be misleading.

Ans. It is no longer figure 1, it is now figure 3. In the opinion of the authors, figure 3 is not misleading because it shows the number of publications, results obtained taking into account the keywords and following our inclusion criteria and exclusion, which are explained in the methodological part (Lines 108-110).

d) The summary of the studies on removal of pollutants represents a lot of work and seems to be well organized. However, conclusions drawn from these tables were often unclear, possibly due to problems in wording.

i) PhCs were referred to as being more/most frequent, but it wasn't clear what that meant. Did these appear in the greatest proportion of publications? Were they in the greatest concentrations?

Ans. PhCs are drugs, likewise, the drugs with the highest frequency of detection mean that they appeared in the highest proportion in the publications.

ii) Different PhCs varied in concentrations from 2 to 6 orders of magnitude from study to study with no means or other type of analyses, so it is not clear to me how these numbers can be compared.

Ans. Regarding the initial concentration of the drugs reported by the articles, these data have been taken only from those studies that were carried out on a real scale, that is, monitoring the concentration of the drugs in the WWTP tributaries, subsequently, we established ranges between data from authors who studied the same drug. It should be noted that comparisons between the studies were not sought, but rather to show the information as reported by the different authors to identify the drugs with the highest detection frequency and those that have been reported in the highest concentrations in wastewater.

iii) For Fig. 8, the claim was made that RO had the highest removal rate, but the way I viewed this graph, it appeared that ozonation and nanofiltration were as effective. This should be clarified. 

Ans. Although it is true that the removal of drugs using reverse osmosis, nanofiltration and ozonation techniques is similar in studies with a quasi-experimental research design, reverse osmosis stood out for achieving a removal of the greatest amount of drugs, eliminating by complete two of these micropollutants. Additionally, the research included a broader discussion of the best removal techniques used in studies with a pre-experimental research design or in large-scale WWTPs.
e) Since this paper seems to take the wide view, looking at both the frequency to which publications appeared and the efficacy of the methods used, there should be one more important point addressed which at best is skimmed over briefly. There is a difference between removal of pollutants and destruction of pollutants. Some methods merely concentrate the pollutants while separating them out from the waste stream; are these the "best" methods if the pollutants still need to be dealt with? Also, any chemical change means that the detection method no longer identifies them. That could be that even 100% "destruction" could still mean a very large amount of potentially bioactive breakdown products are still present. This issue should be mentioned when discussing effectiveness of methods.

Ans. In relation to this topic, the problems of each removal technique are mentioned in the introduction to the manuscript, since currently there is no method that is completely effective.
Regarding the question, the suggestion will be taken into account for the realization of a future investigation.

Round 2

Reviewer 1 Report

The manuscript describes a review of literature about different waste-water treatment systems and focuses the discussion on the drugs' remotion efficiency. The manuscript is pertinent and provides interesting information. Authors have attended almost all the suggestions done during the first review process. However, the response from the authors was not entirely satisfactory:

In the first observation point, where I asked: “Why did the authors not consider including the use of white-rot fungi or enzymatic treatments in their review?”.

The response was not supported adequately, as, indeed, there is a broad literature (including reviews; showing that there exists a lot of information about) that describes the fungal and enzymatic treatment for real (and non-sterile) wastewater, indicating its potential to developing real strategies for drug remotion from real wastewater systems. Authors only recognize the activated sludge (discarding the membrane bioreactors) as the only valid biological treatment system. But as we can find in the literature, treatment systems using fungi or enzymes are considered good alternatives. Even the authors should mention algae-based treatments that, in fact, were considered in the revised papers, but the authors do not say anything about this. I cite some examples below. In that case, probably the authors should consider changing the current title, using the term “conventional techniques” or something like this that be more representative of the scope of the review. And adding a text in the introduction explaining why the authors did not consider these biological methods in the revision; probably, the combinations of keywords did not detect the papers that report these kind of studies. But not to seem as if authors are unaware of these possibilities.

Fungi:

https://doi.org/10.1016/j.cej.2020.128210

https://doi.org/10.1007/s10311-021-01379-5

https://doi.org/10.1016/j.watres.2013.06.007

https://doi.org/10.1016/j.watres.2019.115313

https://doi.org/10.1016/j.watres.2018.02.056

Enzymes:

https://doi.org/10.1016/j.biortech.2021.126201

https://doi.org/10.1007/s40726-021-00183-7

https://doi.org/10.1002/mbo3.722

Algae:

https://doi.org/10.1016/j.watres.2018.03.072

https://doi.org/10.1016/j.ecoenv.2018.12.032

https://doi.org/10.3390/ijerph19137717

https://doi.org/10.1016/j.chemosphere.2021.130674

Table 4 was improved, but still not looking good. I couldn’t read it. Please correct format.

Authors responded in the response letter to what they consider a “pre-experimental design,” “quasi-experimental design,” or “pure experimental design.” They answered, “To place the research design in the tables and figures, the authors were guided by the existing information in the same documents investigated….” But I must insist that these criteria must be specified in the manuscript because readers do not always know exactly which information is in the revised manuscript included in the study.

Author Response

Dear colleague, the authors appreciate the comments made on the work, some of the points have been addressed; Others have been answered from the point of view of the authors and justifying them.
I hope that the manuscript is to your liking.
best regards

The manuscript is pertinent and provides interesting information. Authors have attended almost all the suggestions done during the first review process. However, the response from the authors was not entirely satisfactory:

In the first observation point, where I asked: “Why did the authors not consider including the use of white-rot fungi or enzymatic treatments in their review?”.

The response was not supported adequately, as, indeed, there is a broad literature (including reviews; showing that there exists a lot of information about) that describes the fungal and enzymatic treatment for real (and non-sterile) wastewater, indicating its potential to developing real strategies for drug remotion from real wastewater systems. Authors only recognize the activated sludge (discarding the membrane bioreactors) as the only valid biological treatment system. But as we can find in the literature, treatment systems using fungi or enzymes are considered good alternatives. Even the authors should mention algae-based treatments that, in fact, were considered in the revised papers, but the authors do not say anything about this. I cite some examples below. In that case, probably the authors should consider changing the current title, using the term “conventional techniques” or something like this that be more representative of the scope of the review. And adding a text in the introduction explaining why the authors did not consider these biological methods in the revision; probably, the combinations of keywords did not detect the papers that report these kind of studies. But not to seem as if authors are unaware of these possibilities.

Fungi:

https://doi.org/10.1016/j.cej.2020.128210

https://doi.org/10.1007/s10311-021-01379-5

https://doi.org/10.1016/j.watres.2013.06.007

https://doi.org/10.1016/j.watres.2019.115313

https://doi.org/10.1016/j.watres.2018.02.056

Enzymes:

https://doi.org/10.1016/j.biortech.2021.126201

https://doi.org/10.1007/s40726-021-00183-7

https://doi.org/10.1002/mbo3.722

Algae:

https://doi.org/10.1016/j.watres.2018.03.072

https://doi.org/10.1016/j.ecoenv.2018.12.032

https://doi.org/10.3390/ijerph19137717

https://doi.org/10.1016/j.chemosphere.2021.130674

Ans. The authors consider that the title of the research should not be changed because it would modify the stated objectives and the justification was added for not including treatments that used white rot fungi or enzymatic treatments in the review.

Table 4 was improved, but still not looking good. I couldn’t read it. Please correct format.

Ans. Apparently at the time of submitting the tables and figures they moved, please confirm if they now appear correctly.

Authors responded in the response letter to what they consider a “pre-experimental design,” “quasi-experimental design,” or “pure experimental design.” They answered, “To place the research design in the tables and figures, the authors were guided by the existing information in the same documents investigated….” But I must insist that these criteria must be specified in the manuscript because readers do not always know exactly which information is in the revised manuscript included in the study.

Ans. The explanation regarding the research designs in the methodological part was added to the manuscript.

Reviewer 2 Report

The authors have addressed my comments and suggestions from the revision of the initially submitted document.

For my part, its publication proceeds.

Author Response

Thank you very much dear colleague.
best regards

Reviewer 3 Report

Second round comments

line 131: the paper now clearly states that hybrid and artificial techniques are excluded, but the text fails to state WHY. In responses to reviewers, the authors indicate that they wanted to use only papers disucssing one type of treatment, and that the treatment be of actual wastewater and not an experimental (artificial) setup. Actually saying that in the manuscript would be much more informative than just calling it "exclusionary criteria" which is not informative to the reader.

Figure 1 does not appear to be labeled. It is also rotated 90 degrees in my draft and hard to read. Finally, in the responses to reviewers, there were explanations of what is meant by "quasi-experimental", etc. as well as M20 etc. I was therefore expecting to find that information in the legend for this figure, but I see no legend at all. This information needs to be in the manuscript and associated with this figure, if not in the legend, then in the text.

In my draft, Figure 2 appears twice.

I am not satisfied that one of my original points has been adequately addressed. The data presented are based on measurement of the original molecule in question. Therefore, physical removal of a drug by 100% means that all the pollutant molecules have been removed. However, chemical/biological treatments need only to alter the molecule, and it may no longer be detectable. Therefore what may appear as a 100% removal of a drug may only mean that 100% of the molecules have been modified into some other pollutant that is NOT being measured. Suggesting that the drug has been "eliminated" ignores the possibility of potentially active pollutants still being present even if technically the drug being studied has been "removed". From a "pollution" and "contamination" point of view, this is an important point when evaluating different treatment methods.

Author Response

Dear colleague, I hope you are doing well.
The authors made the changes to the suggested manuscript, with their respective responses to each suggestion.
I hope that the manuscript is to your liking.
best regards

line 131: the paper now clearly states that hybrid and artificial techniques are excluded, but the text fails to state WHY. In responses to reviewers, the authors indicate that they wanted to use only papers disucssing one type of treatment, and that the treatment be of actual wastewater and not an experimental (artificial) setup. Actually saying that in the manuscript would be much more informative than just calling it "exclusionary criteria" which is not informative to the reader.

Ans. Thanks for the comment, a paragraph giving more details about the scope of the research has now been added in the introduction of the manuscript.

Figure 1 does not appear to be labeled. It is also rotated 90 degrees in my draft and hard to read. Finally, in the responses to reviewers, there were explanations of what is meant by "quasi-experimental", etc. as well as M20 etc. I was therefore expecting to find that information in the legend for this figure, but I see no legend at all. This information needs to be in the manuscript and associated with this figure, if not in the legend, then in the text.

Ans. Ok the changes have been made.

In my draft, Figure 2 appears twice.

Ans. The figures were moved at the time of submitting the manuscript, please confirm if they now appear correctly.

I am not satisfied that one of my original points has been adequately addressed. The data presented are based on measurement of the original molecule in question. Therefore, physical removal of a drug by 100% means that all the pollutant molecules have been removed. However, chemical/biological treatments need only to alter the molecule, and it may no longer be detectable. Therefore what may appear as a 100% removal of a drug may only mean that 100% of the molecules have been modified into some other pollutant that is NOT being measured. Suggesting that the drug has been "eliminated" ignores the possibility of potentially active pollutants still being present even if technically the drug being studied has been "removed". From a "pollution" and "contamination" point of view, this is an important point when evaluating different treatment methods.

Ans. A paragraph was added at the end of the discussion of the manuscript, indicating what was requested.
